



**Tracing the evolution of morphology and mixing state of soot**
**particles along with the movement of an Asian dust storm**
Liang Xu[1], Satoshi Fukushima[2], Sophie Sobanska[3], Kotaro Murata[2], Ayumi Naganuma[2], Lei Liu[1],
Yuanyuan Wang[1], Hongya Niu[4], Zongbo Shi[5], Tomoko Kojima[6], Daizhou Zhang[2], Weijun Li[1,*]
[1]Department of Atmospheric Sciences, School of Earth Sciences, Zhejiang University, Hangzhou
310027, China
[2]Faculty of Environmental and Symbiotic Sciences, Prefectural University of Kumamoto,
Kumamoto 862-8502, Japan
[3]Institute of Molecular Sciences, UMR CNRS 5255, University of Bordeaux, 351 cours de la
libération, 33405 Talence, France
[4]Key Laboratory of Resource Exploration Research of Hebei Province, Hebei University of
Engineering, Handan 056038, Hebei, China
[5]School of Geography, Earth and Environmental Sciences, University of Birmingham, UK
[6]Department Earth and Environmental Science, Faculty of Advanced Science and Technology,
Kumamoto University, Kumamoto 860-8555, Japan
*Corresponding author: W. Li (liweijun@zju.edu.cn)
Department of Atmospheric Sciences, School of Earth Sciences, Zhejiang University, Hangzhou
310027, China





**Abstract**
Tracing the aging progress of soot particles during transport is highly challenging. An
Asian dust event could provide an ideal opportunity to trace the continuous aging
progress of long-range transported soot particles. Here, we collected individual aerosol
particles at an inland urban site (T1) and a coastal urban site (T2) in China and a coastal
site (T3) in southwestern Japan during an Asian dust event. Microscopic analysis
showed that the number fraction of soot-bearing particles increased from 19% to 22%
from T1 to T2 in China but surprisingly increased to 56% at T3 in Japan. The dominant
fresh soot (71%) at T1 became partially embedded (70%) at T2 and fully embedded
(84%) at T3. These results indicated that the soot particles had lower deposition than
other aerosol types and became more aged from T1 to T3. The fractal dimension of the
soot particles slightly changed from 1.74 at T1 and 1.78 at T2 but significantly became
1.91 at T3. We found that the soot morphology compressed depending on secondary
coating thickness and relative humidity. Moreover, we observed a unique mixing
structure at T3 that tiny soot particles were seemly broken from large ones cross the
East China Sea and distributed in organic coatings instead of sulfate core in particles.
Our study provide important constraints of the morphological effects to better
understand changes of microscopic structures of soot. These new findings will be
helpful to improve optical calculation and modeling of soot particles and their regional
climate effects in the atmosphere.



## 1. Introduction


Soot (i.e., black carbon (BC)) is a type of carbonaceous material with graphitic
structures emitted from the incomplete combustion of fossil fuels and biomass. Soot
particles exhibit a chain-like aggregation morphology with a diameter of 10 nm to 100
nm (Buseck et al., 2014). Because of its strong capacity to absorb solar radiation, soot
is considered the second greatest contributor to global warming after carbon dioxide
(IPCC., 2013;Bond et al., 2013). Soot is an important particulate pollutant in fine
particles (i.e., $PM_{2.5}$) in urban polluted air, which adversely affects the respiratory health
of citizens and induces generally unwanted heating in the planetary boundary layer
(West et al., 2016;Ding et al., 2016).
Fresh soot particles are hydrophobic but are converted into a hydrophilic state
following their aging through physical and chemical processes (Li et al., 2016b;Riemer
et al., 2010;Perring et al., 2017). Aged soot particles containing secondary coating
aerosols (e.g., ammonium sulfate, ammonium nitrate, and organic matter) can be
activated as cloud condensation nuclei (CCN) (Zhang et al., 2008;Wang et al.,
2010;Ding et al., 2019;Shiraiwa et al., 2007;Lee et al., 2019). These coatings can
significantly change the optical scattering and absorption capacity of soot particles (Liu
et al., 2017;Moffet and Prather, 2009;Matsui et al., 2018;He et al., 2015;Zhang et al.,
2018a). Numerical model simulations have estimated that light absorption by internally
mixed soot is enhanced by a factor of 2 over externally mixed soot (Jacobson, 2001).
In contrast, Cappa et al. (2012) reported in situ observations of soot absorption
enhancement of only 6% in ambient air. This discrepancy between simulation and





observation could be attributed to the complex mixing structure and various
morphologies of soot particles in the air (Adachi et al., 2016;Li et al., 2016a;Wu et al.,

2018).

In aged air masses, soot particles tend to be internally mixed with secondary

aerosols such as sulfates, nitrates, and secondary organic matter (Li et al., 2016b).
Especially in the East Asian region, one of the most polluted areas in the world, soot is
internally mixed with secondary aerosols in polluted urban, rural, and remote air
(Adachi et al., 2016;Zhang et al., 2013;Yuan et al., 2019;Zhang et al., 2018b). However,
most of these studies have focused on the aging and mixing state of soot particles at one
or multiple isolated sites. These results have not traced the detailed aging processes
(e.g., morphology and mixing structure) from fresh to aged soot particles during their
transport.

Although great progress has been made in the field of soot aging, it is highly

challenging to trace the aging processes of soot particles during transport. Asian dust
storms carry both dust and anthropogenic aerosols across East Asia into the North
Pacific Ocean (Li et al., 2014;Geng et al., 2014;Zhang et al., 2005). This presents an
ideal environment to study the aging processes of soot particles during long-range
transport. Compared to previous publications, the present study quantified the variation
in mixing structures and fractal dimension of soot particles and further explored how
shape of soot particles changed following the dust storm movement from East China to
Japan.

Using transmission electron microscopy (TEM), we investigated the morphology,



mixing structure, relative abundance, and size distribution of individual soot particles.
Furthermore, we evaluated the morphological differences of individual soot particles at
three sampling sites. Finally, a conceptual model was proposed to better understand the
aging processes of long-rang transported soot particles.
**2. Experimental methods**
**2.1 Aerosol sampling**
Three sampling sites were chosen for aerosol collections: an inland urban site in
Jinan city (T1, 36.67°N 117.06°E), China, a coastal urban site in Qingdao city (T2,
36.10°N 120.46°E), China, and a coastal rural site at Amakusa (T3, 32.30°N 130.00°E)
in southwestern Japan (Figure 1). A dust storm outbreak was observed in East Asia.
Detailed information about this dust storm will be discussed in Section 3.1. We
collected aerosol particles during dust transport from 18 to 19 March 2014 at the three
sampling sites (Figure S1-S5). In total, seven dust samples were collected within 30
hours after the dust storm arrival. The details about the sampling dates, times,
meteorological conditions, and PM (particulate matter) concentrations for the samples
are listed in Table S1.
A DKL-2 sampler (Genstar Electronic Technology, China) was used to collect
individual aerosol particles on copper TEM grids covered by carbon film (carbon type-
B, 300-mesh copper; Tianld Co., China) with an air flow of 1.0 L/min. A single-stage
impactor with a 0.5 mm diameter jet nozzle was installed on the sampler. This impactor
has a collection efficiency of 100% at an aerodynamic diameter of 0.5 μm with an
assumed particle density of 2 g/cm$^3$. The sampling duration varied from 1 min to 10





min according to the visibility, PM concentration, and particle distribution on the
substrate. All samples were placed in sealed, dry plastic capsules and stored in a
desiccator at 25 °C and 20 ± 3% relative humidity (RH) for further analysis.
**2.2 Electron microscopic analyses**

A JEOL JEM-2100 transmission electron microscope (TEM) operated at 200 kV

was used to analyze individual particles. Elemental composition was determined
semiquantitatively by using an energy-dispersive X-ray spectrometer (EDS) (Oxford
Instruments, UK) that can detect elements heavier than carbon ($Z \geq 6$). The distribution
of aerosol particles on TEM grids was not uniform, with coarser particles occurring
near the center and finer particles occurring on the periphery (Xu et al., 2019). Therefore,
to ensure that the analyzed particles were representative of the entire size range, three
areas were chosen from the center to the periphery of the sampling spot on each grid.
iTEM software (Olympus Soft Imaging Solutions GmbH, Germany) was used to
analyze the TEM images and obtain the projected area, perimeter, aspect ratio, and
equivalent circle diameter (ECD) of individual aerosol particles. In total, we analyzed
412, 486, and 887 aerosol particles for T1, T2, and T3 site, respectively.
**2.3 AFM analysis**

Atomic force microscopy (AFM) is an analytical method used for studying the

surface structure of solid materials. AFM (Dimension Icon, Germany) can determine
the three-dimensional morphology of particles in tapping mode. The AFM settings
consisted of imaging forces between 1 and 1.5 nN, scanning rates between 0.5 and 0.8
Hz, and a scanning range of 10 μm with a resolution of 512 pixels per length. The



bearing areas (A) and bearing volumes (V) of the particles were directly obtained from
NanoScope Analysis software. Their equivalent circle diameters (ECDs) and equivalent
volume diameters (EVDs) were calculated according to the formulas described by Chi
et al. (2015).
The correlations of ECDs and EVDs are shown in Figure S6 in the Supporting
Information. Therefore, the ECD of individual aerosol particles measured from the
iTEM software can be further converted into an EVD based on this correlation.
**2.4 Air mass backward trajectories**
Forty-eight hour backward trajectories were calculated for the three sites using the
NOAA HYSPLIT (Hybrid Single Particle Lagrangian Integrated Trajectory) trajectory
model (Stein et al., 2015). We selected an altitude of 1500 m as the end point in each
backward trajectory.
We measured the actual duration from the Beijing-Tianjin-Hebei (BTH) area to T1
and T2 according to the backward trajectories in Figure 1. It was approximately 12
hours between BTH and T1 and 15 hours between BTH and T2. The interval between
T1 and T2 was three hours. The duration between the air mass leaving T2 and reaching
T3 was approximately 30 hours.
**2.5 Morphological analysis of soot particles**
The fractal dimension ($D_f$) calculated by the scaling law is used to characterize the
morphology of soot particles (Koeylue et al., 1995).
$$N = k_g \left( \frac{2R_g}{d_p} \right)^{D_f} \tag{1}$$

where N is the total number of soot monomers, $R_g$ is the radius of gyration of the soot



particle, $d_p$ is the diameter of soot monomer, $k_g$ is the fractal prefactor, and $D_f$ is the
mass fractal dimension of an individual soot particle.

$D_f$ and $k_g$ in Equation 1 are estimated from a power law fit of a scatter plot of N

versus the values of $2R_g/d_p$. N can also be calculated by Equation 2.
$$N=k_a\left(\frac{A_a}{A_p}\right)^{\alpha}$$     (2)
where $A_a$ is the projected area of the soot particle, $A_p$ is the mean projected area of the
soot monomer, $k_a$ is a constant, and $\alpha$ is an empirical projected area exponent.

The values of $\alpha$ and $k_a$ in Equation 2 depend on the overlap parameter ($\delta$) calculated

using Equation 3 (Oh and Sorensen, 1997).
$$\delta=\frac{2a}{l}$$     (3)
where a is the soot monomer radius and l is the monomer spacing.

The radius of gyration of the soot particle $R_g$ is obtained by the simple

correlation in Equation 4 developed by Brasil et al. (1999)
$$L_{max}/(2R_g)=1.50 \pm 0.05$$     (4)
where $L_{max}$ is the maximum length of the soot particle.

The values of $d_p$, $A_a$, $A_p$, a, l, and $L_{max}$ can be directly obtained from TEM images.

In addition to $D_f$, we also used the aspect ratio (AR) to further quantify the

roundness of soot particles. The aspect ratio is the maximum ratio between the length
and width of a bounding box (Equation 5). An aspect ratio of 1 (the lowest value)
indicates that a particle is not elongated in any direction.
$$AR=\frac{L_{max}}{W_{max}}$$     (5)
where $L_{max}$ is the maximum length of a soot particle and $W_{max}$ is the maximum width



of a soot particle.

## 3. Results and discussion

### 3.1 The Asian dust storm event

Figure 2 displays variations in $PM_{10}$ and $PM_{2.5}$ concentrations before, during, and
after the dust storm event at the Jinan, Qingdao, and Amakusa sampling sites. The dust
storm air mass started to influence T1 at approximately 14:00 on 03/17 (BJT, Beijing
Time, UTC+8). The concentration of $PM_{10}$ at T1 increased rapidly to a maximum value
of 834 μg/m$^3$. The air mass reached T2 at 17:00 on 03/17, and the highest $PM_{10}$
concentration was recorded at 721 μg/m$^3$. After the arrival of a cold front at T2, the air
mass continued moving approximately 1000 km to T3 at 17:00 on 03/18. The
concentration of $PM_{10}$ reached 87 μg/m$^3$ at T3 (Figure 2). During this study, the
meteorological data (e.g., temperature and air pressure) measured at the three sampling
sites also confirm the arrival time of the dust storm (Figures S3-S5). All seven dust
samples were collected after the arrival of the dust storm, thus confirming the sampling
of the same dust storm event (Figures 1b and S2).
Figure 1 indicates that all the air masses during the dust storm event originated
from Mongolia, moving southeastward via the BTH area, reaching T1 and T2,
respectively, within a 3-hour interval. The BTH, as the largest city cluster in China,
contains one of the largest anthropogenic emission sources (e.g., heavy industries, coal-
fired power plants, and vehicles) in the world (Li et al., 2016b). The ground PM and
meteorological measurements at the three sampling sites (Figure 2 and S3-S5) coupled
with air mass back trajectories (Figure 1) and a dust storm simulation in East Asia
(Figure S1) together verified that the dust storm event, under the force of a strong cold
front, transported across the large BTH city cluster to the downwind area. Therefore,



this dust storm movement provides a unique opportunity to study particles in the same
air mass and thus trace physical and chemical changes in aerosol particles.
**3.2 Classification and mixing state of soot-bearing particles**
Soot particles with a typical chain-like structure can be easily distinguished from
other aerosol components (e.g., sulfate, organic, metal, and mineral particles) by their
morphology. TEM observation is a convenient way to determine whether soot is
associated with other aerosol components (Li et al., 2016b;Laskin et al., 2019). During
the dust storm period, 56% of the analyzed particles within a size range of 50 nm to 2.4
μm included soot particles at T3, approximately three times higher than those at T1
(19%) and T2 (22%). This high percentage of internally mixed soot particles was also
shown by Ueda et al. (2016) in an Asian outflow at Noto Peninsula, Japan, based on
single-particle soot photometer (SP2) analyses. Our results show that the dust storm
event not only carried large amounts of dust particles from the Gobi Desert in
northwestern China but that this dust-laden air mass also incorporated many soot
particles from polluted East Asia (Figure 2 and Figure 3a-d). This is consistent with Pan
et al. (2015), who showed that dust storms in East Asia contain and transport
anthropogenic pollutants from urban areas.
Based on the mixing structures between soot and sulfate on the substrates, three
groups of soot particles were defined in this study: fresh, partially embedded and fully
embedded (Figure 3).
*Fresh soot*. The soot particles were not obviously mixed with secondary aerosol
components (Figure 3a). Although surfaces of the fresh soot particles could contain
minor organic matter, the organic film was insufficient to change soot morphology and
optical properties (Buseck et al., 2014).





*Partially embedded soot*. Part of the soot particle was coated by secondary aerosols
(Figure 3b).
*Fully embedded soot*. The entire soot particle was encapsulated by secondary
aerosols (Figure 3c). We also noticed that some soot particles were only embedded in
the organic coating instead of the sulfur-rich core (Figure 3d).
TEM images show that the fully embedded soot particles with a clear rim on the
substrate displayed a droplet-like shape (Figure 3c-d), suggesting that these secondary
particles were in an aqueous phase in ambient air (Li et al., 2016b).
Based on the three mixing structures of soot particles, we further obtained their
relative abundance at the three sampling sites (Figure 4). Seventy-one percent of soot-
bearing particles were fresh at T1, decreasing to 16% at T2. In contrast, partially
embedded soot increased from 14% at T1 to 70% at T2 when the cold front moved from
T1 to T2. It should be noted that fresh soot disappeared at T3 after crossing the East
China Sea, and the fully embedded soot dominated soot-bearing particles (84%).
Following the dust storm movement, we found that the number fraction of total
soot-bearing particles increased to 56% among all the analyzed particles from T1 to T3,
suggesting that soot particles had lower deposition than other aerosol types in the cold
front. Indeed, soot particles normally have smaller sizes and densities than mineral dust,
metal, sulfate, and nitrate particles (Peng et al., 2017), suggesting that soot particles can
be transported over longer distances during Asian dust storms.
**3.3 Quantifying the morphology of soot particles**
The fractal dimension ($D_f$) of soot particles is a key parameter used to reflect soot


morphological structure; e.g., compact soot particles usually have larger $D_f$ than lacy
aggregates (China et al., 2015;Wang et al., 2017;China et al., 2013). Therefore, $D_f$ can
be used to understand soot aging processes in the atmosphere. Figure 5a shows that the
$D_f$ sequence of soot particles is T1 (1.74 ± 0.10) < T2 (1.78 ± 0.16) < T3 (1.91 ± 0.04).
The $D_f$ of soot particles at T1 and T2 (1.74 ± 0.10 and 1.78 ± 0.16) is much closer to
the values of soot emitted from sources, such as the $D_f$ from biomass burning in the
range of 1.68−1.74 (Chakrabarty et al., 2006) and the $D_f$ from diesel burning in the
range of 1.56−1.68 (Wentzel et al., 2003). The $D_f$ of soot particles at T3 (1.91 ± 0.04)
is similar to that of aged soot (1.81-1.90) in remote marine air (China et al., 2015) and
polluted air in North China (Wang et al., 2017).

At the three sampling sites, the highest $D_f$ value at T3 suggests a more compacted

structure of the soot particles. Moreover, we obtained the aspect ratios of soot particles
at the three sampling sites, which can indicate the roundness of the particle shape (Yuan
et al., 2019). The average aspect ratio of soot particles at T3 was 1.56, much lower than
1.72 at T1 and 1.66 at T2 (Figure 5b). These two parameters show that the soot
morphology became more compact and had a rounder shape following the dust storm
movement.
**3.4 Soot-bearing particle size growth following soot aging**

The average ratio ($D_p/D_{core}$) of the diameter of the internally mixed particle ($D_p$) to

its corresponding soot core ($D_{core}$) during the dust storm period was 1.42 at T1, 1.78 at
T2, and 2.49 at T3 (Figure 5b). The $D_p/D_{core}$ values in this study are much higher than
the reported values in fresh emissions (e.g., average value 1.24 for fossil fuel (Sahu et
al., 2012)) but close to ~2.0 in aged aerosols in background and polluted air (Dahlkötter




et al., 2014;Raatikainen et al., 2015;Metcalf et al., 2012). Recently, Peng et al. (2017)
reported a high growth rate in urban Beijing and a derived average $D_p/D_{core}$ value of
1.97 (1.34-2.61). The $D_p/D_{core}$ value in urban Beijing air is much higher than our
reported values of 1.42-1.78 at T1 and T2 during the dust storm period. This is
understandable considering the weak secondary aerosol formation in the dust storm in
the continental air as a result of acidic gases being scavenged by the large amounts of
mineral dust particles (Li et al., 2016b).

Based on the air mass backward trajectories, we can infer that it took approximately

three hours for the cold front to move between T1 and T2 and 30 hours from T2 to T3
(Figure 1). Here, we calculated the coating volume of aged soot particles based on the
values of $D_p$ and $D_{core}$ of individual particles and found a 152% increase in the coating
volume from T1 to T2 and a 609% increase from T2 to T3.
**3.5 Aging mechanism of soot particles**

We noticed that the partially embedded soot particles significantly increased from

14% at T1 to 70% at T2 (Figure 4), indicating that the fresh soot particles aged during
the dust storm movement from the inland to the coastal area. However, we found that
the $D_f$ value at T1 only slightly changed from 1.74 at T1 to 1.78 at T2. These results
indicate that the morphological structures of soot particles underwent slight changes,
although large amounts of fresh soot converted into partially embedded soot particles
from T1 to T2.

Figure 4 shows that the fresh soot particles disappeared at T3, and the number

fraction of fully embedded soot particles increased to 84%. Moreover, the $D_f$ of soot
particles had a large change from 1.78 at T2 to 1.91 at T3, which suggests that the
morphology structure of soot particles changed from chain-like to compact when the
air masses crossed the East China Sea (Figure 5a). This large change in soot

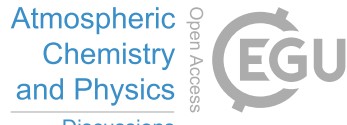

morphology from T2 to T3 is different from the slight change in soot particles from T1
to T2. The contrasting results suggest that soot particles underwent more complicated
aging processes in marine air than in continental air.

Secondary aerosol formation on soot particles can significantly change their fractal

morphology into a compact shape (China et al., 2015;Wang et al., 2017;Ma et al.,
2013;Pei et al., 2018). The thick coating of soot particles occurred when air masses
crossed the East China Sea (Figure 5b), suggesting that secondary aerosol coating
formation can significantly compress the fractal morphology of soot particles. Recently,
Yuan et al. (2019) further found that the phase change of secondary aerosols (due to RH
variation) in aged soot particles could further compress the fractal shape of soot
aggregates. The high humidity in marine air (T2 to T3) should lead to phase changes of
secondary aerosols and further cause the morphological compactness of soot
aggregations. These two reasons can explain why the large change in soot fractal
dimension occurred from T2 to T3 instead of T1 to T2 (Figure 5a).

TEM observations present a particular mixing structure of the fully embedded soot

at T3: organic coating instead of sulfate contains several typical soot particles, and the
organic coating spreads on the substrate (named droplet-like particles (O'Brien et al.,
2015;Li et al., 2011)) (Figure 3d). The droplet-like coating morphology of soot can
reflect that these secondary particles were in an aqueous phase at T3 in the air. A
previous study has shown that secondary aerosol particles begin to acquire aqueous
shells at RH 60% (Sun et al., 2018). Once secondary aerosols change from a solid to
liquid phase following an RH increase in marine air, soot particles tend to adhere to the
liquid phase through coagulation (Li et al., 2016b). Figure 3d shows the phase
separation of the organic coating and sulfate core on the substrate under the
phenomenon of liquid-liquid phase separations (You et al., 2012). Recently,



Brunamonti et al. (2015) found that soot particles tend to redistribute into the organic
coating during liquid-liquid phase separation. Therefore, the soot distribution in the
organic coating indicates that aerosol particles in the air mass at T3 underwent an
aqueous aging process over the East China Sea, which is different from the continental
aerosol particles at T1 and T2. It must be noted that several tiny soot particles were
distributed in the organic coating at T3 (Figure 3d), which did not occur at T1 and T2.
Our findings suggest that the complex cloud-aqueous process of individual particles in
marine air could result in scattered soot particles.
Tracing the soot particles during the dust storm, we can clarify that the morphology
change of soot particles depends on the secondary coating thickness and relative
humidity in the air. Moreover, the cloud-aqueous process and the phase separation of
organic and sulfate components in the soot-bearing particles likely break the chain-like
soot and change soot distribution within individual secondary aerosol particles. The
microscopic changes between soot and coating could change their optical absorption,
which is different from the core-shell absorption (He et al., 2015). Our study proposes
that BC-related optical models should not only consider the mixing state of soot
particles but also incorporate the morphological structure of soot particles in different
environmental air.
Based on the results and discussion above, we propose a conceptual model to
summarize the evolution of morphology and mixing state of soot particles along with
the movement of an Asian dust storm (Figure 6). Dust storms in East Asia could carry
soot and other anthropogenic pollutants from urban areas to downwind areas. During
the transport, the dominated mixing structure of individual soot particles changed from
fresh to partially embedded and finally to fully embedded. Meanwhile, the chain-like
soot compressed and had a rounder shape depending on secondary coating thickness





and relative humidity.

**4. Conclusions**

Individual aerosol particles were collected from 18 to 19 March 2014 during an

Asian dust storm event. Three sampling sites along with the pathway of the dust storm
were chosen to study soot aging, including an inland urban site in Jinan city, China (T1),
a coastal urban site in Qingdao city, China (T2), and a coastal rural site at Amakusa in
southwestern Japan (T3). Soot-bearing particles were classified into three types: fresh,
partially embedded, and fully embedded. There was a noticeable difference in the
mixing structure of soot particles during long-range transport, with 71% fresh soot in
the analyzed soot particles (by number) at T1, 70% partially embedded soot at T2, and
84% fully embedded soot at T3. The fractal dimension ($D_f$) of soot particles at T3 (1.91)
was higher than that at the other two sites (1.74 and 1.78), suggesting that soot particles
converted from chain-like to compact shapes during long-range transport. This study
showed that an increasing number of soot particles were internally mixed with
secondary aerosol particles and significantly aged during transport. The average ratio
of $D_p/D_{core}$ during the dust storm period was 1.42 at T1, 1.78 at T2, and 2.49 at T3,
indicating increasing coating thickness. By comparing the soot fractal dimension in
continental air and marine air, we found that secondary coating thickness and relative
humidity both can significantly change the fractal morphology of soot particles in the
air. Individual particle analysis showed that several tiny soot particles only observed in
organic coatings instead of sulfate in individual soot-bearing particles at T3, suggesting



that the complicated aging processes of individual particles can break the chain-like
soot formation.

**Data availability**
All data presented in this paper are available upon request from the corresponding
author (liweijun@zju.edu.cn).
**Supporting information**
Table S1 and Figures S1-S6
**Author contributions**
LX and WL conceived the study and wrote the manuscript. The field campaign was
organized and supervised by WL and DZ. SF, KM, AN, and TK collected aerosol
particles. LX, SS, LL, YW, HN, and ZS contributed sample and data analyses. All
authors reviewed and commented on the paper.
**Competing interests**
The authors declare that they have no conflict of interest.
**Acknowledgments**
We thank Peter Hyde for his editorial comments. This work was funded by the National Key R&D
Program of China (2017YFC0212700), the National Natural Science Foundation of China
(91844301, 41807305), Zhejiang Provincial Natural Science Foundation of China (LZ19D050001),
and China Postdoctoral Science Foundation (2019M662021).

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

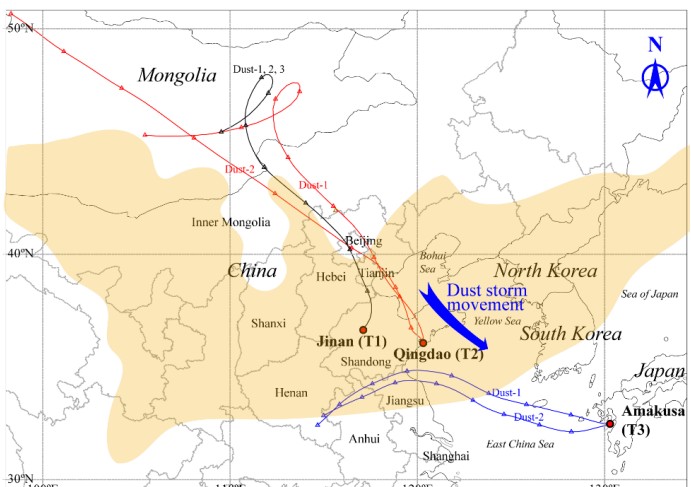


Figure 1. The locations of the three sampling sites and HYSPLIT forty-eight hour air
mass backward trajectories arriving at 1500 m above ground level at T1, T2, and T3
sites. The interval between two triangle symbols is six hours. The yellow shadow is
derived from Figure S1, which represents the area influenced by the dust storm at 08:00
on 2014/03/18 (BJT, UTC+8).

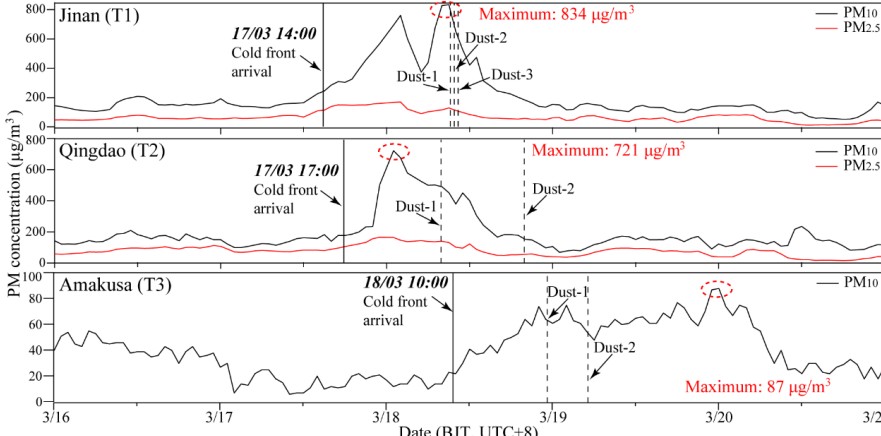


Figure 2. Time series of PM (particulate matter) concentrations at T1, T2, and T3 during
sampling. Data sources: T1 and T2: The Ministry of Ecology and Environment of the
People's Republic of China, https://www.aqistudy.cn/; T3: National Institute for
Environmental Studies of Japan, https://www.nies.go.jp/igreen/).





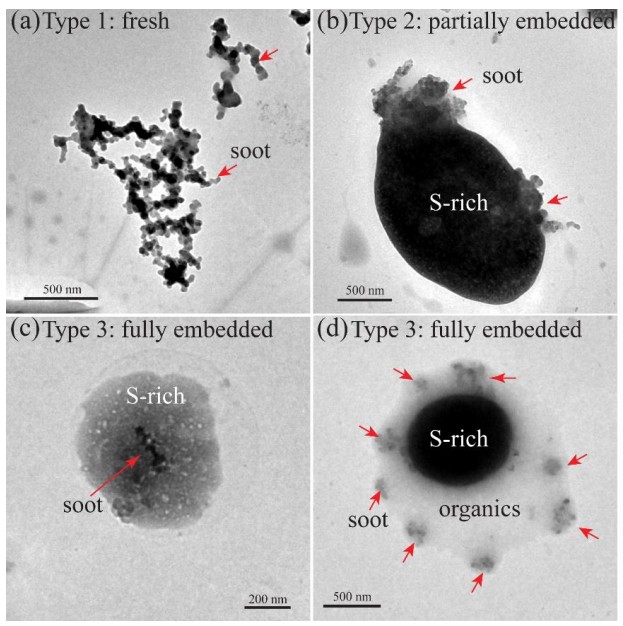

Figure 3. Morphology and relative abundance of soot-bearing aerosol particles: (a) fresh chain-like soot aggregates with no visible coating; (b) partially embedded soot: part of the soot particle was coated by secondary aerosols; (c) fully embedded soot: the whole soot particle was encapsulated by secondary aerosols; (d) a subtype of fully embedded soot: individual soot particles were only embedded in the organic coating on a sulfur-rich particle.

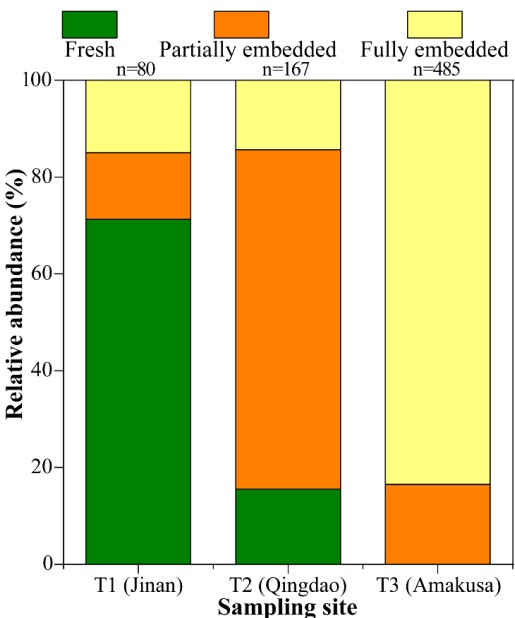

Figure 4. Relative abundance of three types of soot-bearing aerosol particles at the three

sampling sites. The number of analyzed soot-bearing particles is shown above the

column.

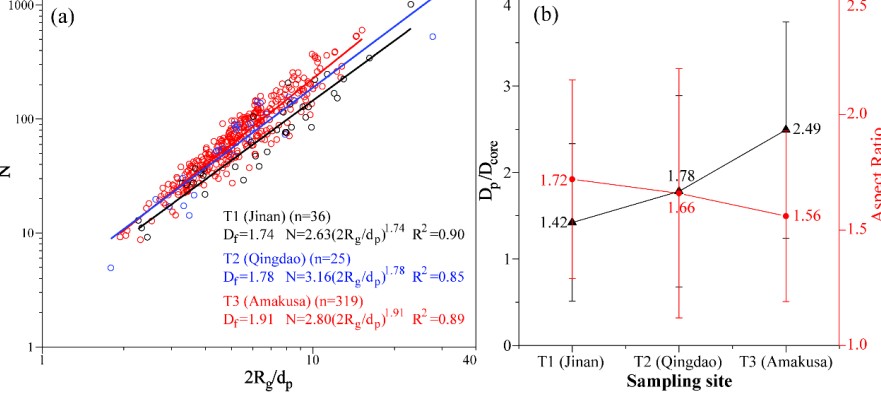

Figure 5. (a) Fractal dimension of soot-bearing particles at the three sampling sites. The

parameter n in parentheses represents the total number of soot particles analyzed for

each site to calculate $D_f$ and $k_g$. (b) The particle-to-soot core diameter ratio ($D_p/D_{core}$)

and aspect ratio of soot-bearing particles at the three sampling sites.





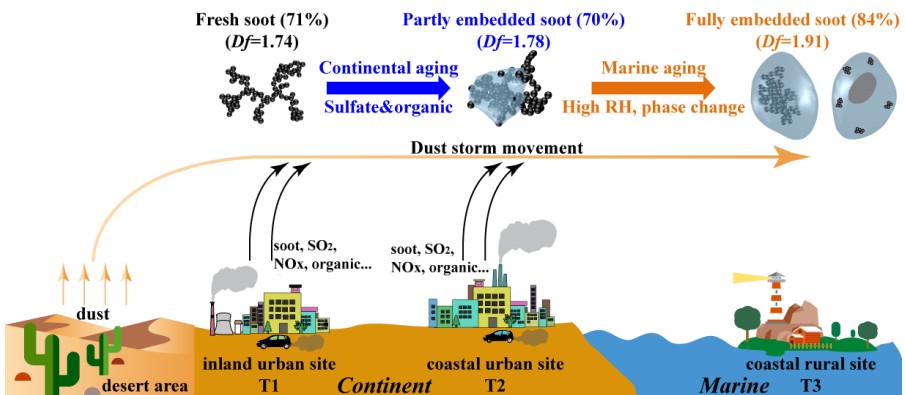

Figure 6. Schematic diagram showing the evolution of morphology and mixing state of soot particles along with the movement of an Asian dust storm