# Peer review of "Tracing the evolution of morphology and mixing state of soot 1 particles along with the movement of an Asian dust storm 2 Liang Xu1, Satoshi Fukushima2, Sophie Sobanska3, Kotaro Murata2, Ayumi Naganuma2, Lei Liu1, 3 Yuanyuan Wang1, Hongya Niu4, Zongbo Shi5, Tomoko Kojima6, Daizhou Zhang2, Weijun Li1,\* 4 5 1Department of Atmospheric Sciences, School of Earth Sciences, Zhejiang University, Hangzhou"

_Atmospheric Chemistry and Physics, 2020_

## Referee Comment (RC1) · Anonymous Referee #1 · 20 Jul 2020

This study investigates soot mixing states mainly using a transmission electron microscope (TEM) in China and Japan. They collected samples by following the aging of an air plume during a dust event and found changes of soot mixing states as the plume aged. Their findings will be an interesting case study to follow the soot aging in an ambient atmosphere and a heavily polluted area. As the soot aging process is of interest for an understanding of their climate influences through the accurate estimates of soot lifetime and optical properties, their result will have contributions to the climate prediction. However, I have a list of concerns regarding their discussion and interpretations of their results. A major concern is that the many discussion regarding T3 samples (and Type 3 particle) was based only on a particle shown in Fig. 3 d. The authors also

try to have a general model based on the one-particle observation; a better statistic is necessary here. Please see the specific comments for more suggestions.

1. Line 108-110: from 1 min to 10 min: I see only 1 to 2 min in Table S1.

2. Line 113 (and 86): TEM has been defined as "transmission electron microscopy" in line 86, whereas it is defined as "transmission electron microscope" here. Please be consistent.

3. Section 2.5 Line 160-162: It is unclear how the Eq. 3 is used to obtain $\alpha$ and ka here.

4. Line 180-183: How were the cold front arrival times defined? I do not see a clear change at the time of the arrival in Figures S3-5. As the timing is used to evaluate soot aging time, please make them specific. Why did the authors use Beijing time instead of UTC? As the local time in T3 site is not Beijing time, UTC will be better.

5. Line 196-198: The dust distribution and back-trajectory model in Fig. 1 indicates that the Dust samples at T3 site are not from the dust plume. On 8:00 18 March in Fig. 1, the edge of the dust plume was 18 and 24 hours ahead for the Dust 1 and 2 sampling at T3 site, respectively. Thus, the dust should arrive between 2:00-8:00 19 March at T3 site, whereas the samplings were done at 23:16, 18 March and 6:36, 19 March. At least, the sentence "together verified that the dust storm event" is not valid, unless I miss something. Figure S2 may be helpful, but I cannot see details in the figure. I recommend rechecking the modeling data and showing robust data to prove the dust arrival before the sampling at T3 site. TEM data of dust particles from the T3 site samples may be useful.

6. Line 205-208: It is surprising that the dust samples have so many soot particles. Are there so many dust particles as well together with soot particles? TEM images with low-magnification showing more particles, including soot and dust, will be helpful to have an idea of how such a large number of soot particles occur in the dust samples.

7. Line 223-224: Do coagulations between soot particles and others contribute to the "partially embedded soot" formation?

8. Line 237-242: When the soot particles are fully coated, such as aged samples at T3, the deposition efficiency of fully embedded soot-bearing particles may not differ from those without soot. I assume that the increase of the number fraction of soot-bearing particles is simply due to an increase of mixing state index as aging (Riemer and West, 2013; Healy et al., 2014), but more discussion will be useful here.

9. Line 247-254 and 258-259: I suggest adding TEM images, including many soot particles having high or low fractal dimensions, so that readers visually see the compaction of soot particles as age.

10. Line 263-265: It is unclear how the Dcore was obtained. As soot particles have a fractal structure, Dp and Dcore should have different relations in Fig. S6. Please explain the methodology.

11. Line 307-327: The discussion in this paragraph based on only one particle (Fig. 3d), and it is difficult to have a general conclusion. I suggest showing more particle images or adding statistics of the fraction having particles with phase separation or tiny soot particles. I also question how the fractal dimensions or other parameters were obtained from the scattered soot particles.

12. Line 326-330: I do not see any evidence of cloud-aqueous process discussed here. Are there any could in the dust plume? How do "the cloud-aqueous process and the phase separation of organic and sulfate components in the soot-bearing particles likely break the chain-like soot" happen? Do the "the cloud-aqueous process and the phase separation" make soot compact or scatter? Careful discussions should be provided here.

13. Line 363-366: The conclusion was obtained only one soot particle. Please show more data or revise the conclusion. It sounds that the conclusion "that the complicated aging processes of individual particles can break the chain-like soot formation" contradicts with soot compaction process. Please have a comment on this.

14. Figure 2: Dust 1 in T2 was collected ahead of the samples in T1. Dust 2 in the T2 was collected only 3 hour ahead of the T3 dust 1 sample. Ideally, if someone wants to compare the aging process, the samples should be collected within the same plume, following the airmass movement. The sampling strategy may cause an influence, and some discussion about the sampling strategy may be helpful.

15. Figure 3: I recommend having more particle images, including low-magnification images from T1, T2, and T3 sites.

16. Figures 4 and 5a. The number of analyzed soot particles does not agree between Fig. 4 and 5a (e.g., 80 vs 36). Did the author select soot particles for the analysis in Fig. 5a? If so, how and does the method cause any bias?

17. Fig. 5b: The error bars are too high to compare the data. What do the bars indicate?

18. Fig. 6: If there are soot and other emissions at T2, the aging process will be rather complicated, i.e., they should be the mixtures of aged soot and freshly emitted soot. The influence of the emission at T2 should be discussed.

19. Table S1. The wind directions for the samples within the same sampling site vary largely (almost opposite directions (e.g., 121 vs. 358 at T3). The wind direction in Fig S1 at T3 is also complicated. I question if the samples were collected from dust plume. Again, TEM images of dust particles may be helpful.

20. Figure S2. It is difficult to see the details.

21. Figure S6. Is the plot include soot particles?

Healy, R. M., Riemer, N., Wenger, J. C., Murphy, M., West, M., Poulain, L., Wiedensohler, A., O'Connor, I. P., McGillicuddy, E., Sodeau, J. R., and Evans, G. J.: Single

particle diversity and mixing state measurements, Atmos. Chem. Phys., 14, 6289–6299, https://doi.org/10.5194/acp-14-6289-2014, 2014.

Riemer, N. and West, M.: Quantifying aerosol mixing state with entropy and diversity measures, Atmos. Chem. Phys., 13, 11423–11439, https://doi.org/10.5194/acp-13-11423-2013, 2013.

---

## Referee Comment (RC2) · Anonymous Referee #2 · 3 Aug 2020

The study by Xu et al. investigates the aging of soot particles during Asian dust event. The authors collected samples at three sites and compared soot aging at single particle basis. They mainly used transmission electron microscopy (TEM) to study morphology of soot particles. Several morphological descriptors such as aspect ratio, fractal dimensions are used to quantify morphology of soot particles and they classified mixing state of soot-bearing particles based on their coating thickness. They found that soot particles are compact with highest fractal dimension of soot collected at coastal site (T3) in southwestern Japan. They suggested that compact morphology is due to condensation of secondary coating material and though phase separation at high humidity during transport. The research topic is certainly relevant, but I have several concerns.
There is not much discussion about the chemical composition of soot-bearing particles. Did you perform any chemical analysis?

The backtrajectory analysis is not properly discussed in the manuscript. However, this is key for the dust event discussion. I didn't follow the cold front arrival. The authors should discuss in more detail.

What is the relevance of dust storm here? If soot particles are studied during a dust storm, why the authors didn't observe or discuss about mixing of dust and soot particles. The authors should discuss about number fraction of soot particles that are mixed with dust particles and size distribution of both dust and soot particles.

The authors should discuss about the size distribution of three types of soot particles. How did the authors calculate fractal dimension of partially embedded and fully embedded soot particles? For type 3, especially for the fragmented soot ones, it is difficult to measure the required parameters to calculate fractal dimension. They should also provide fractal dimension separately for all three types of soot.

The discussion about the mixing state configuration needs to be elaborated, like how many soot particles did you observe within individual partially embedded soot particles? How many fragments were observed in type-3 (figure 3d). This information would be useful to understand the aging process and for modeling purposes.

Need to discuss how many soot particles were studied per sample. How many total samples during event? Overall, particle statistic is poor.

If phase separation may be a key mechanism for observation of fully embedded particles, the authors can investigate fraction of fully embedded particles at different RH, not just by sampling site. The RH was high too during certain time at T2 site. The authors should investigate those samples as well. Interactive comment

---

## Author Comment (AC1) · 3 Oct 2020

**General Response: We thank the Referee for your helpful comments. We have addressed all comments and provided point by point response below. The revised manuscript is presented in below Response**

This study investigates soot mixing states mainly using a transmission electron microscope (TEM) in China and Japan. They collected samples by following the aging of an air plume during a dust event and found changes of soot mixing states as the plume aged. Their findings will be an interesting case study to follow the soot aging in an ambient atmosphere and a heavily polluted area. As the soot aging process is of interest for an understanding of their climate influences through the accurate estimates of soot lifetime and optical properties, their result will have contributions to the climate prediction. However, I have a list of concerns regarding their discussion and interpretations of their results. A major concern is that the many discussion regarding T3 samples (and Type 3 particle) was based only on a particle shown in Fig. 3 d. The authors also try to have a general model based on the one-particle observation; a better statistic is necessary here. Please see the specific comments for more suggestions.

Answer: We appreciated the Referee#1's comments which significantly improve the quality of the manuscript. We carefully answer them one by one as below. The modifications were highlighted in red in the revised manuscript.

1. Line 108-110: from 1 min to 10 min: I see only 1 to 2 min in Table S1.

Answer: Sorry for the misunderstanding. The "1 min to 10 min" is the usual collection time for samples in our overall campaign, but for the dust event samples, the duration is "1 to 2 min". This mistake is corrected in the manuscript.

2. Line 113 (and 86): TEM has been defined as "transmission electron microscopy" in line 86, whereas it is defined as "transmission electron microscope" here. Please be consistent.

Answer: The "transmission electron microscope" has been revised to "transmission electron microscopy".

3. Section 2.5 Line 160-162: It is unclear how the Eq. 3 is used to obtain α and ka here.

Answer: Eq. 3 is used to obtain the overlap parameter (δ). Then the overlap parameter (δ) can be used to acquire α and ka based on Figure 6. in Oh and Sorensen (1997). More detailed information was added in the revised manuscript: "The values of α and ka in Equation 2 depend on the overlap parameter (δ) calculated using Equation 3. Then δ can be used to obtain α and ka based on Fig. 6. in Oh and Sorensen (1997).".

[Figure]

**FIG. 6.** The parameters $\alpha$ and $k_a$ of Eq. [12] versus overlap parameter for three different fit ranges in $N$, the number of monomers per cluster.

Figure cited from Oh and Sorensen (1997)

4. Line 180-183: How were the cold front arrival times defined? I do not see a clear change at the time of the arrival in Figures S3-5. As the timing is used to evaluate soot aging time, please make them specific. Why did the authors use Beijing time instead of UTC? As the local time in T3 site is not Beijing time, UTC will be better.

Answer: The local time in T3 site is Japan Standard Time, which is one hour ahead of Beijing Time. In the previous manuscript, we have transformed Japan Standard Time to Beijing Time at T3 to facilitate comparison. We also realize that UTC is much better as the Referee#1 suggested. Therefore, all time used in this paper has been revised to UTC.

The cold front arrival times were defined based on PM concentration in Figure 2 and meteorological data (mainly pressure) in Figures S3-S5. The air behind a cold front is colder and drier than the air in front. When the cold front passes through, the RH and temperature can drop, and the pressure can increase. A clear turning point of pressure is observed in T1 and T2 (Figures S3 and S4), thus we consider this point as the arrival time. For T3, the turning point of pressure is not obvious, but based on the RH and temperature dropping, we still can confirm the arrival time.

We also want to note that the cold front arrival times is not used to evaluate soot aging time, it is only used to confirm the time when the sampling site starts to be influenced by the dust storm. The aging time is obtained from the backward trajectories in Figure 1.

5. Line 196-198: The dust distribution and back-trajectory model in Fig. 1 indicates that the Dust samples at T3 site are not from the dust plume. On 8:00 18 March in Fig. 1, the edge of the dust plume was 18 and 24 hours ahead for the Dust 1 and 2 sampling at T3 site, respectively. Thus, the dust should arrive between 2:00-8:00 19 March at T3 site, whereas the samplings were done at 23:16, 18 March and 6:36, 19 March. At least, the sentence "together verified that the dust storm event" is not valid, unless I miss something. Figure S2 may be helpful, but I cannot see details in the figure. I recommend rechecking the modeling data and showing robust data to prove the dust arrival before the sampling at T3 site. TEM data of dust particles from the T3 site samples may be useful.

Answer: There might be some misunderstanding to interpret the Figure 1. We cannot define the arrival time based on the dust plume (yellow shadow) and backward trajectories. In fact, the intersections between the dust plume and backward trajectories are meaningless. The Dust 1 and 2 samplings were done at 15:16 and 22:36, 18 March (UTC). Therefore, 18 and 24 hours ahead of Dust 1 and 2 is 21:16 and 22:36, 17 March. This is even not the time of the dust plume we present in Figure 1 (00:00, 18 March).

The cold front arrival time was defined based on PM concentration and meteorological data as mentioned in Comment #4.

[Figure]

6. Line 205-208: It is surprising that the dust samples have so many soot particles. Are there so many dust particles as well together with soot particles? TEM images with low-magnification showing more particles, including soot and dust, will be helpful to have an idea of how such a large number of soot particles occur in the dust samples.

Answer: We found some but not many dust particles that mixed with soot particles in T3 site. A low-magnification TEM image of T3 is added in the manuscript (Figure 7).

[Figure]

Figure 7. Low-magnification TEM images at T1, T2, and T3.

7. Line 223-224: Do coagulations between soot particles and others contribute to the "partially embedded soot" formation?
Answer: Yes, it is.

8. Line 237-242: When the soot particles are fully coated, such as aged samples at T3, the deposition efficiency of fully embedded soot-bearing particles may not differ from those without soot. I assume that the increase of the number fraction of soot-bearing particles is simply due to an increase of mixing state index as aging (Riemer and West, 2013; Healy et al., 2014), but more discussion will be useful here.
Answer: Thanks for the valuable suggestion. We add the discussion that the increase of mixing state index during soot aging could lead to the number fraction increase (Line 247-251): "Moreover, the number fraction increase of soot-bearing particles also could be attributed to the increase of mixing state index (the metric to quantify the population mixing state, ranging from 0 for a completely external mixture to 1 for a completely internal mixture) as aging during transport (Riemer and West, 2013;Healy et al., 2014)."

9. Line 247-254 and 258-259: I suggest adding TEM images, including many soot particles having high or low fractal dimensions, so that readers visually see the compaction of soot particles as age.
Answer: The $D_f$ in this paper was acquired using statistical method (slope in Figure 5), the method cannot provide $D_f$ value for one single soot particle. Here, we provide model simulated soot particles with different $D_f$ to represent different soot morphology. The new figure should be helpful for the readers to visually understand the soot compaction.

[Figure]

Figure 5. Fractal dimension of soot-bearing particles at the three sampling sites. The parameter n in parentheses represents the total number of soot particles analyzed for each site to calculate Df and kg.

10. Line 263-265: It is unclear how the Dcore was obtained. As soot particles have a fractal structure, Dp and Dcore should have different relations in Fig. S6. Please explain the methodology.

Answer: For soot particles with fractal shape, we manually draw an interpolated polygon to fully cover the edge of soot particles (see figures below). Our iTEM software could obtain the area of this polygon based on our manual drawing, and further convert the area to the equivalent circle diameter (ECD) of soot particles.

[Figure]

[Figure]

The relations of ECD and EVD in Figure S6 are obtained for sulfur-rich particles. During sample collection, the impaction could lead to morphological change (see figure below). The particle ECD from TEM is larger than its diameter in the air. Thus, we convert S-rich particles' ECD to EVD to acquire the Dp.

But the size of soot is less affected by the impaction. We apply the ECD of soot as the Dcore.

[Figure]

About the conversion from ECD to EVD, our previous studies have quantify the secondary sulfate particles (Chen et al., 2017; Zhang et al., 2020). The same method was adopted in this study.

11. Line 307-327: The discussion in this paragraph based on only one particle (Fig. 3d), and it is difficult to have a general conclusion. I suggest showing more particle images or adding statistics of the fraction having particles with phase separation or tiny soot particles. I also question how the fractal dimensions or other parameters were obtained from the scattered soot particles.

Answer: We add the low-magnification TEM images of T1, T2, and T3 as suggested in Comment #15. More particles with phase separation and tiny soot are shown in the T3 image

(Figure 7).

[Figure]

Figure 7. Low-magnification TEM images at T1, T2, and T3.

The fractal dimensions and other parameters of the scattered soot particles were acquired same as other soot particles. We manually draw a interpolated polygon to fully cover the edge of soot particles and obtain data from our iTEM software.

[Figure]

12. Line 326-330: I do not see any evidence of cloud-aqueous process discussed here. Are there any cloud in the dust plume? How do "the cloud-aqueous process and the phase separation of organic and sulfate components in the soot-bearing particles likely break the chain-like soot" happen? Do the "the cloud-aqueous process and the phase separation" make soot compact or scatter? Careful discussions should be provided here.

Answer: We agree with the referee that the expression in this sentence is not precise. Especially we do not have solid proof for the cloud process.

The aqueous process could be a possible reason for the formation of scattered soot in coating. Secondary aerosol particles begin to acquire aqueous shells at RH 60% (Sun et al.,

2018). The high RH in marine air could lead to secondary aerosols change from a solid to liquid phase. Therefore, we propose that the aqueous process and the phase separation of organic and sulfate likely is the reason of soot in the coating.

As for the tiny size soot in the coating, there is no previous studies to report the scattered tiny soot phenomenon. Here we proposed possible reasons for this: (1) the aqueous process and the phase separation break the chain-like soot; (2) soot particle with smaller size have a longer lifetime and could be transported over longer distances. From our current knowledge, the second reason has a better chance to explain this. Thus, we revise the corresponding part in the manuscript (Line 337-341): "There is no previous study to report the tiny scattered soot in the organic coating. We proposed a possible reason that soot particle with smaller size have a longer lifetime and could be transported over longer distances. Therefore, the tiny soot particles have more chances to coagulate with preexisting aqueous secondary particles in marine air (Liu et al., 2018)." However, the reason for this requires further work.

13. Line 363-366: The conclusion was obtained only one soot particle. Please show more data or revise the conclusion. It sounds that the conclusion "that the complicated aging processes of individual particles can break the chain-like soot formation" contradicts with soot compaction process. Please have a comment on this.
Answer: Thanks. We seriously consider this comment here. TEM image of particles with phase separation and tiny soot are shown in the T3 image.
The contradicted sentence was deleted from the conclusion to avoid potential misunderstanding.

[Figure]

Figure 7. Low-magnification TEM images at T1, T2, and T3.

14. Figure 2: Dust 1 in T2 was collected ahead of the samples in T1. Dust 2 in the T2 was collected only 3 hour ahead of the T3 dust 1 sample. Ideally, if someone wants to compare the aging process, the samples should be collected within the same plume, following the airmass movement. The sampling strategy may cause an influence, and some discussion about the sampling strategy may be helpful.
Answer: Yes, this is the most ideal way to observe the aging process. However we cannot know the precise movement of dust air mass during in-situ sample collection. The influence of dust air mass could last a relatively long time at one site. Therefore, we adopted all the samples collected in the period under the influence of the dust air mass (Figure 2). Thus, even though the samples are not precisely sampled following the air mass movement, the samples could still

represent the characteristics of soot aging. As our prepared study and clear aim, this is the best way to achieve this study.

15. Figure 3: I recommend having more particle images, including low-magnification images from T1, T2, and T3 sites.
Answer: Low-magnification images from T1, T2, and T3 were added in the manuscript (Figure 7).

[Figure]

Figure 7. Low-magnification TEM images at T1, T2, and T3.

16. Figures 4 and 5a. The number of analyzed soot particles does not agree between Fig. 4 and 5a (e.g., 80 vs 36). Did the author select soot particles for the analysis in Fig. 5a? If so, how and does the method cause any bias?
Answer: Actually, we did not select soot particles for the $D_f$ analysis in Figure 5. There are some soot particles that can be recognized as soot structure but they are not clear enough to provide necessary data for $D_f$ analysis (mainly in the low-magnification TEM images).
For example, the figure below is not clear any more for $D_f$ analysis, but we can identify this soot from their EDS and morphology. The information still can be used for the statistic in Figure 4.

[Figure]

As a result, the soot number in Figure 4 and Figure 5 is not consistent. We can guarantee that we do not occasionally chose soot particles in the sample.
For better description, we add more explanations in the caption of Figure 5: "The inconsistency

of analyzed soot number in Figure 4 and 5 is attributed to the indistinct soot particles in the low-magnification TEM images that can be identified as soot but cannot provide necessary data for Df analysis".

17. Fig. 5b: The error bars are too high to compare the data. What do the bars indicate?

Answer: The error bars indicate the standard deviation. We realize that the average values and error bars are not sufficient to compare the data. Therefore, we replace it with the box plot to clearly show the variation of data. Also, we replace previous aspect ratio with a commonly used morphological descriptor shape factor to better represent soot morphology difference.

[Figure]

Figure 6. (a) Shape factor of soot-bearing particles at the three sampling sites and (b) the particle-to-soot core diameter ratio (Dp/Dcore).

18. Fig. 6: If there are soot and other emissions at T2, the aging process will be rather complicated, i.e., they should be the mixtures of aged soot and freshly emitted soot. The influence of the emission at T2 should be discussed.

Answer: We agree with the referee that local emissions at T2 could influence the observation of soot aging process. In our study, dust samples at T2 were all collected during the dust storm period. The strong diffusion during the dust storm is not conducive to soot accumulation. Thus, long-range transported soot is still the dominant at T2 among the dust period.

We add more discussion in Line 295-298: "The strong diffusion during the dust storm is not conducive to soot accumulation (Pan et al., 2015). Although local emissions at T2 could interference the observation of soot aging process, long-range transported soot particles were still dominant at T2 during the cold front."

19. Table S1. The wind directions for the samples within the same sampling site vary largely (almost opposite directions (e.g., 121 vs. 358 at T3). The wind direction in Fig S1 at T3 is also complicated. I question if the samples were collected from dust plume. Again, TEM images of dust particles may be helpful.

Answer: The wind directions recorded during our sampling could be considered as the instant wind direction (two minutes maximum, Table S1). This instant direction cannot represent the regional wind. However, the PM concentration, meteorological data, and the dust distribution could provide solid proof for the sample collection in dust plume.

20. Figure S2. It is difficult to see the details.

Answer: We adjusted the arrangement of Figure S2 to make the details more readable.

[Figure]

21. Figure S6. Is the plot include soot particles?

Answer: As mentioned in Comment #10, the relations of ECD and EVD in Figure S6 are obtained for sulfur-rich particles.

References:

Healy, R. M., Riemer, N., Wenger, J. C., Murphy, M., West, M., Poulain, L., Wiedensohler, A., O'Connor, I. P., McGillicuddy, E., Sodeau, J. R., and Evans, G. J.: Single particle diversity and mixing state measurements, Atmos. Chem. Phys., 14, 6289–6299, https://doi.org/10.5194/acp-14-6289-2014, 2014.

Riemer, N. and West, M.: Quantifying aerosol mixing state with entropy and diversity measures, Atmos. Chem. Phys., 13, 11423–11439, https://doi.org/10.5194/acp-13-11423-2013, 2013.

Chen, S., Xu, L., Zhang, Y., Chen, B., Wang, X., Zhang, X., et al. (2017). Direct observations of organic aerosols in common wintertime hazes in North China: insights into direct emissions from Chinese residential stoves. *Atmospheric Chemistry and Physics*, *17*(2), 1259-1270. https://doi.org/10.5194/acp-17-1259-2017

Oh, C., & Sorensen, C. M. (1997). The Effect of Overlap between Monomers on the Determination of Fractal Cluster Morphology. *Journal of Colloid and Interface Science*, *193*(1), 17-25. https://doi.org/10.1006/jcis.1997.5046

Sun, J., Liu, L., Xu, L., Wang, Y., Wu, Z., Hu, M., et al. (2018). Key Role of Nitrate in Phase Transitions of Urban Particles: Implications of Important Reactive Surfaces for Secondary Aerosol Formation. *Journal of Geophysical Research: Atmospheres*, *123*(2), 1234-1243. https://doi.org/10.1002/2017JD027264

Zhang, J., Liu, L., Xu, L., Lin, Q., Zhao, H., Wang, Z., et al. (2020). Exploring wintertime regional haze in northeast China: role of coal and biomass burning. *Atmospheric Chemistry and Physics*, *20*(9), 5355-5372. https://doi.org/10.5194/acp-20-5355-2020

---

## Author Comment (AC2) · 3 Oct 2020

**General Response: We thank the Referee for your helpful comments. We have addressed all comments and provided point by point response below. The revised manuscript is presented in below Response**

The study by Xu et al. investigates the aging of soot particles during Asian dust event. The authors collected samples at three sites and compared soot aging at single particle basis. They mainly used transmission electron microscopy (TEM) to study morphology of soot particles. Several morphological descriptors such as aspect ratio, fractal dimensions are used to quantify morphology of soot particles and they classified mixing state of soot-bearing particles based on their coating thickness. They found that soot particles are compact with highest fractal dimension of soot collected at coastal site (T3) in southwestern Japan. They suggested that compact morphology is due to condensation of secondary coating material and though phase separation at high humidity during transport. The research topic is certainly relevant, but I have several concerns.

Answer: We appreciated the Referee#2's comments which significantly improve the quality of the manuscript. We carefully answer them one by one as below. The modifications were highlighted in red in the revised manuscript.

1. There is not much discussion about the chemical composition of soot-bearing particles. Did you perform any chemical analysis?

Answer: Yes, we applied chemical analysis using an energy-dispersive X-ray spectrometer (EDS). The sulfur-rich and organics are the main component mixed with soot, this has been mentioned in 3.2 and Figure 3. Because soot have typical morphology and only contain C and minor O, therefore we did not have more discussion on it.

2. The backtrajectory analysis is not properly discussed in the manuscript. However, this is key for the dust event discussion. I didn't follow the cold front arrival. The authors should discuss in more detail.

Answer: (1) More discussion about the backward trajectory is added in the manuscript (Line 195-198): "The transport duration from the BTH to T1 and T2 was about 12 hours and 15 hours, respectively. Thus, we estimated that the interval between T1 and T2 was three hours. After passing over T1 and T2, the air masses kept moving southeastward to Japan. The estimated interval between T2 and T3 was 30 hours."

(2) The cold front arrival times is used to confirm the time when the sampling site starts to be influenced by the dust storm. They were defined based on PM concentration in Figure 2 and meteorological data (mainly pressure) in Figures S3-S5. Here we add more description in caption of Figure 2: "The cold front arrival times indicate the time when the sampling site starts to be influenced by the dust storm."

3. What is the relevance of dust storm here? If soot particles are studied during a dust storm, why the authors didn't observe or discuss about mixing of dust and soot particles. The authors should discuss about number fraction of soot particles that are mixed with dust particles and size distribution of both dust and soot particles.

Answer: We appreciate the referee's comments. Soot particles mostly have smaller size <500

nm but dust particles mostly are in larger size (> 2um). Although we observed several dust particles associated with soot particles, the number faction is too small (Figure 7). That is the reason that we did not describe more about the mixture of dust and soot particles. Here we mainly consider the movement of the strong cold front during the dust storm. Here the dramatic high PM10 concentrations during the dust storm can be helpful to confirm the cold front. In light of the primary purpose of this study, we did not focus on the dust particles like our previous studies such as (Li et al., 2016;Li and Shao, 2009).

The size distribution of soot particles is presented in the next comment.

[Figure]

Figure 7. Low-magnification TEM images at T1, T2, and T3.

4. The authors should discuss about the size distribution of three types of soot particles. How did the authors calculate fractal dimension of partially embedded and fully embedded soot particles? For type 3, especially for the fragmented soot ones, it is difficult to measure the required parameters to calculate fractal dimension. They should also provide fractal dimension separately for all three types of soot.

Answer: As the referee's comments, we add more data as below.

(1) The size distribution of soot particles at three sites is provided as follow (Line 273-276): "Size distribution of the soot core indicates a small difference between T1, T2, and T3 during the dust storm period (Figure S9). Thus, the Dp/Dcore increase from T1 to T3 is attributed to the increased coating thickness."

[Figure]

Figure S8. Size distribution of soot core (exclude coating) at T1, T2, and T3.

(2) For soot particles with fractal shape, we manually draw a interpolated polygon to fully cover the edge of soot particles (see figures below). Our iTEM software could obtain the area of this polygon based on our manual drawing, and further convert the area to the equivalent circle diameter (ECD) of soot particles.

[Figure]

[Figure]

The fractal dimensions and other parameters of the scattered soot particles were acquired same as other soot particles. We manually draw a interpolated polygon to fully cover the edge of soot particles and obtain data from our iTEM software.

[Figure]

It is true that measuring the required parameters to calculate fractal dimension is difficult, especially in low-magnification TEM images. But, we still can acquire the corresponding parameters from high-resolution TEM images.

[Figure]

(3) Thanks to Referee's valuable advice, we realized that there was an issue in the $D_f$ analyses. Thus, we recalculated the $D_f$ in Figure 5. However, there are few soot particles at T1 and T2 that can be used for $D_f$ analyses (34 and 21,respectively). We cannnot provide fractal dimension of all three types of soot at T1 and T2. As for T3, we calculate the fractal dimension of partially and fully embeded in the Supporting Information (no fresh soot particle observed at T3).

[Figure]

Figure 5. Fractal dimension of soot-bearing particles at the three sampling sites. The parameter n in parentheses represents the total number of soot particles analyzed for each site to calculate Df and kg.

[Figure]

Figure S7. Fractal dimension of partially and fully embedded soot particles at T3 site. It is different to provide fractal dimension of different types of soot at T1 and T2 because of the small number of soot particles at these two sites. The parameter n in parentheses represents the total number of soot particles analyzed for each site to calculate $D_f$ and $k_g$

5. The discussion about the mixing state configuration needs to be elaborated, like how many soot particles did you observe within individual partially embedded soot particles? How many fragments were observed in type-3 (figure 3d). This information would be useful to understand the aging process and for modeling purposes.

Answer: We provided the frequency of soot fragment number in single soot-bearing particles. More discussion is added in Line 228-229: "Most of partially embedded soot particles include one soot core, only ~ 10% of them contain two soot cores (Figure S7)";

Line 321-323: "More than half of this type of particles contain one or two soot fragments, while 43% of them include more than three soot fragments (Figure S7)."

[Figure]

Figure S7. Frequency of soot fragment number in single soot-bearing particles.

6. Need to discuss how many soot particles were studied per sample. How many total samples during event? Overall, particle statistic is poor.

Answer: Totally, we analyzed seven dust samples (Table S1). The number of analyzed soot-bearing particles is shown above the column in Figure 4. The total number of aerosol particles analyzed in this study is presented in the 2.2, which is 412, 486, and 887 for T1, T2, and T3 site, respectively.

[Figure]

Figure 4. Relative abundance of three types of soot-bearing aerosol particles at the three sampling sites. The number of analyzed soot-bearing particles is shown above the column.

7. If phase separation may be a key mechanism for observation of fully embedded particles, the authors can investigate fraction of fully embedded particles at different RH, not just by sampling site. The RH was high too during certain time at T2 site. The authors should investigate those samples as well.

Answer: There might be some misunderstanding. We did not propose that phase separation may be a key mechanism for fully embedded particles. The high RH in marine air could lead to secondary aerosols change from a solid to liquid phase. During or after sampling, due to the RH decreasing, the phase separation between inorganic and organic components occur. This is a possible cause of the formation of scattered tiny soot in the organic coating (special mixing structure in Figure 3d).

**References**

Li, W., Sun, J., Xu, L., Shi, Z., Riemer, N., Sun, Y., Fu, P., Zhang, J., Lin, Y., Wang, X., Shao, L., Chen, J., Zhang, X., Wang, Z., and Wang, W.: A conceptual framework for mixing structures in individual aerosol particles, J. Geophys. Res.: Atmos., 121, 13784-13798, https://doi.org/10.1002/2016JD025252, 2016.

Li, W. J., and Shao, L. Y.: Observation of nitrate coatings on atmospheric mineral dust particles, Atmos. Chem. Phys., 9, 1863-1871, https://doi.org/10.5194/acp-9-1863-2009, 2009.

---

## Author Response (AR2)

**General Response: We thank the Referee for your helpful comments. We have addressed**

**the comments and provided response below. The revised manuscript is presented in below**

**Response**

Thank you for the careful consideration of the reviewers' comments. The manuscript has been largely improved. The low-magnification TEM image (Fig. 7) helps understand the samples.

Although the figure is useful, I recommend showing more soot particles in all TEM images, i.e., Fig. 7 (a) has no fresh soot image, and Fig 7(b) only has one soot particles. Overall, the revised manuscript looks well.

Answer: We appreciated the Referee#1's comments which significantly improve the quality of the manuscript. We provided more 
[revised manuscript text omitted]